# Surface Properties of *Parabacteroides distasonis* and Impacts of Stress-Induced Molecules on Its Surface Adhesion and Biofilm Formation Capacities

**DOI:** 10.3390/microorganisms9081602

**Published:** 2021-07-27

**Authors:** Jordan Chamarande, Lisiane Cunat, Céline Caillet, Laurence Mathieu, Jérôme F. L. Duval, Alain Lozniewski, Jean-Pol Frippiat, Corentine Alauzet, Catherine Cailliez-Grimal

**Affiliations:** 1SIMPA, Université de Lorraine, F-54000 Nancy, France; jordan.chamarande@univ-lorraine.fr (J.C.); lisiane.cunat@univ-lorraine.fr (L.C.); alain.lozniewski@univ-lorraine.fr (A.L.); jean-pol.frippiat@univ-lorraine.fr (J.-P.F.); corentine.alauzet@univ-lorraine.fr (C.A.); 2CNRS, LIEC, Université de Lorraine, F-54000 Nancy, France; celine.caillet@univ-lorraine.fr (C.C.); jerome.duval@univ-lorraine.fr (J.F.L.D.); 3Ecole Pratique des Hautes Etudes (EPHE), Laboratoire de Chimie Physique et Microbiologie pour les Matériaux et l’Environnement (LCPME), Paris Sciences Lettres University (PSL), F-54500 Nancy, France; laurence.mathieu@univ-lorraine.fr; 4CHRU de Nancy, Service de Microbiologie, F-54000 Nancy, France

**Keywords:** *Parabacteroides distasonis*, stress, gut microbiota, adhesion and biofilm capacities

## Abstract

The gut microbiota is a complex and dynamic ecosystem whose balance and homeostasis are essential to the host’s well-being and whose composition can be critically affected by various factors, including host stress. *Parabacteroides distasonis* causes well-known beneficial roles for its host, but is negatively impacted by stress. However, the mechanisms explaining its maintenance in the gut have not yet been explored, in particular its capacities to adhere onto (bio)surfaces, form biofilms and the way its physicochemical surface properties are affected by stressing conditions. In this paper, we reported adhesion and biofilm formation capacities of 14 unrelated strains of *P. distasonis* using a steam-based washing procedure, and the electrokinetic features of its surface. Results evidenced an important inter-strain variability for all experiments including the response to stress hormones. In fact, stress-induced molecules significantly impact *P. distasonis* adhesion and biofilm formation capacities in 35% and 23% of assays, respectively. This study not only provides basic data on the adhesion and biofilm formation capacities of *P. distasonis* to abiotic substrates but also paves the way for further research on how stress-molecules could be implicated in *P. distasonis* maintenance within the gut microbiota, which is a prerequisite for designing efficient solutions to optimize its survival within gut environment.

## 1. Introduction

A vast array of microbes, mostly bacteria, colonize the human body [1]. Most of these bacteria are located in the intestine as part of the gut microbiota (GM), with microbiota levels between 10^5^ bacteria per gram of digesta in the upper parts of the small intestine, and more than 10^12^ bacteria per gram of digesta in the large intestine [2]. Several of these bacteria are merely opportunistic colonizers, while the majority are true symbiotic organisms [3]. The human GM consists primarily of the two phyla, *Firmicutes* and *Bacteroidetes*, as well as *Actinobacteria*, *Proteobacteria*, *Fusobacteria* and *Verrucomicrobia* [4]. The majority of these taxa have essential physiological functions such as breaking down fibers, stimulating the immune system or preventing pathogen colonization and are also involved in many other processes [5]. Intestinal dysbiosis, a pathological alteration of GM composition, has been associated with the onset or exacerbation of several chronic disorders such as inflammatory bowel diseases, metabolic syndromes, neurodegenerative and psychiatric conditions [6]. The composition of GM can be influenced by many factors, including intrinsic (host genetics, hormones) and extrinsic (diet, antibiotics, pollutants, and many others) factors [7]. An increasing number of studies has also shown that even moderate can lead to dysbiosis either directly via the action of stress mediators, released in the lumen, or indirectly by modulating the intestinal environment (local immunity, intestinal motility, pH) [3,8,9].

In our previous studies using two distinct murine models of chronic stress, we observed a stress-induced decrease in the intra-caecal relative abundance of *Parabacteroides*, and more specifically of *P. distasonis*, a Gram-negative strictly anaerobe belonging to the *Tannerellaceae* family within the *Bacteroidetes* phylum [10,11]. In spite of its low abundance, this genus was described as a member of the human and mouse core microbiomes [12,13,14]. This can be explained by its small but specific repertoire of degrading enzymes making it a specialist fibrolytic bacterium [15,16]. Moreover, some strains of *P. distasonis* have been shown to possess anti-inflammatory properties both in vitro and in vivo and to reduce the severity of intestinal inflammation in murine models of colitis [17,18,19,20]. As a result of the production of succinate and secondary bile acids, this species also attenuates tumorigenesis and improves metabolic disorders [18,21,22]. Although there are several studies pointing to *P. distasonis* benefits and its potential use as the next-generation of probiotics [20,23], none of them have yet explored the mechanisms that may explain the persistence or loss of *P. distasonis* in the gut. Such mechanisms are related to bacterial surface properties leading to aggregation, adhesion and/or biofilm formation, that could all potentially be influenced by stress markers released by the host.

The predominant mechanism for bacteria to colonize an environment is the formation of a biofilm. Development of a biofilm on the surface, biotic or abiotic, results from both continuous deposition of planktonic cells and bacterial growth. Bacterial colonization begins through a series of recruitment processes that lead to several stages in the biofilm formation. The very first ones are clearly governed by physicochemical processes with the adhesion of planktonic bacteria to surface by the weak van der Waals and London dispersion forces and hydrophobic interactions. The colonist can then anchor to the surface using cell adhesion structure such as pili or fimbriae. At this point, the attachment that was non-specific and reversible become specific and irreversible. The biofilm, that can be made of one or several microorganisms, will then grow by cell division, cell recruitment and exopolysaccharide (EPS) matrix synthesis. Numerous functions have been attributed to EPS, most of them related to protection against the external environment or to the retain of water and nutrients. The EPS matrix is mainly composed of polysaccharides, structural proteins, enzymes, nucleic acids and lipids. Then, the biofilm enters its last stage corresponding to its degradation and dispersion that allow its propagation [24,25,26]. For a given species to form such a bacterial community, it must have the required adhesion capability which strongly depends on the physicochemical cell surface properties (electrostatic charge, presence of adhesins, degree of hydrophobicity) and the surrounding physicochemical conditions (pH, nature of electrolyte ions, salinity, etc). During a host stress situation, catecholamines (epinephrine, norepinephrine and dopamine), glucocorticoids (cortisol) and other stress-induced molecules (such as serotonin) are synthetized and released into blood circulation, oral cavity and gut lumen. It has been reported that these molecules can impact both in vitro and in situ the physiological behavior of bacteria, including their growth features, adhesion and/or biofilm formation capacities [27,28,29,30,31,32]. With the general objective to better understand the maintenance of this beneficial bacterial species within the GM, we characterized in this work the homo-aggregation, adhesion and biofilm formation capacities of 14 unrelated strains of *P. distasonis* and examine their response to stress molecules. Colloidal cell stability against aggregation was addressed via macroscopic homo-aggregation testing method [33,34,35,36]. In the literature, the terminology ‘auto-aggregation’ is sometimes used. However, this latter term tacitly implies that cell aggregation is mediated by auto-associative adhesins located at the surface of the interacting cells [37], rather than by conventional screening of the surface charge that cells carry, leading to suppression of inter-bacteria repulsive interactions. As the exact mechanism of aggregation is unknown for *P. distasonis*, we thus employ the more neutral term ‘homo-aggregation’ taken from text-book colloidal stability literature. In order to determine the charge density and permeability of their external surfaces, analyses of the electrokinetic cell surface properties were performed using electrophoresis.

## 2. Materials and Method

### 2.1. Bacterial Strains and Culture Conditions

A total of 14 *P. distasonis* strains, including 13 nonredundant clinical and clonally unrelated (as assessed by AP-PCR and ERIC-PCR [38,39]; see Appendix A) isolates collected by the clinical microbiology laboratory of the University Hospital of Nancy, France, and the type strain *P. distasonis* DSM 20701^T^ were tested.

*Lactobacillus rhamnosus* GG ATCC 53103 (LGG) was used as positive control for the homo-aggregation and adhesion capacity [40]. *Bacteroides fragilis* 638R (*Bf* 638R) and *Bacteroides thetaiotaomicron* VPI-5482 (*Bt* VPI-5482) were respectively used as positive and negative control for the biofilm formation study [41,42].

Strains were stored at −80 °C in appropriate broth supplemented with 15% (*v/v*) glycerol. Prior to assay, *P. distasonis* strains, *B. fragilis* 638R and *B. thetaiotaomicron* VPI-5482 strains were streaked onto Brucella agar plates (BBA) supplemented with 5% (*v/v*) of defibrinated sheep blood (Oxoid, Thermo Fisher Diagnostics, Dardilly, France), 1% (*v/v*) of hemin (Sigma-Aldrich, Saint Quentin Fallavier, France) and 1% (*v/v*) of vitamin K1 (Sigma-Aldrich). After an incubation at 37 °C for 48 h in anaerobic conditions, strains were subcultured in Schaedler broth for 24 h in anaerobic conditions. LGG was streaked onto MRS agar and incubated at 37 °C for 24 h in aerobic conditions. Further subculture was performed in Schaedler broth for 24 h in aerobic conditions.

### 2.2. Sample Purity and Bacterial Enumeration by Quantitative Polymerase Chain Reaction (qPCR)

qPCR was used for sample purity and bacterial enumeration in the adhesion and biofilm formation assays. In all assays, genomic DNA was extracted by using the QiaAmp DNA MiniKit (Qiagen, Courtaboeuf, France). The extracted DNA was added to 25 µL of the qPCR mixture containing Mesa Blue (2X, Master Mix Plus for SYBR, Eurogentec, Seraing, Belgium) and the following primers (10 μM): primers targeting an internal fragment of panbacterial 16S rRNA encoding gene (1369F: 5′-CGGTGAATACGTTCCCGG-3′ and 1492R: 5′-TACGGYTACCTTGTTACGACTT-3′) versus 16S rRNA-gene-targeted species-specific primers for *P. distasonis* (Pdist1115F: 5′-CTTGCCACTAGTTACTAACA-3′ and Pdist1276R reverse: 5′-CCCTGTCGCCAGGTG-3′) as previously described [10]. For DNA amplification, the following thermocycling conditions were applied with the MyiQ™2 real-time PCR system (Bio-Rad Laboratories, Marnes-la-Coquette, France): initial denaturation at 95 °C for 5 min followed by 40 cycles of 95 °C for 15 s and 60 °C (annealing and extension temperature) for 1 min.

### 2.3. Survivability of P. distasonis under Experimental Conditions

Three media were tested for the survivability of *P. distasonis* DSM 20701^T^ under aerobic or anaerobic conditions: saline solution (NaCl 0.9%), ten times-dilutedMueller-Hinton (MH, Sigma-Aldrich) and 1–250 mM NaNO_3_ (Sigma-Aldrich) solution. The saline solution is used for bacterial suspension dilutions, diluted MH for adhesion capacity assays and NaNO_3_ electrolyte for electrokinetic experiments as a dispersing medium. In experiments conducted in saline solution and diluted MH medium, bacterial suspensions (10^8^ CFU/mL) were incubated for 48 h at room temperature under aerobic conditions and at 37 °C under anaerobic conditions, respectively. After 2, 4, 6, 24 and 48 h, an enumeration on BBA medium was carried out. An enumeration of *P. distasonis* DSM 20701^T^ was also performed after 1 h of incubation at room temperature under aerobic conditions in 1 mM and 250 mM NaNO_3_ electrolyte solution. Two independent experiments were performed including triplicate of each conditions.

### 2.4. Homo-Aggregation Assays

The homo-aggregation property of *P. distasonis* was evaluated (two independent experiments, in triplicate) as previously described by Polak-Berecka et al. [40]. From colonies obtained on plates, a bacterial suspension of 10^8^ CFU/mL was prepared in saline solution. The optical density (*OD_600nm_*) has been measured on a PV4 spectrophotometer (VWR, Fontenay-sous-Bois, France) using polystyrene cuvettes. *OD_600nm_* was then recorded after 0, 2, 3 and 5 h of incubation at room temperature. The ‘homo-aggregation coefficient time’ (*ACT*) was estimated on the basis of the following Equation (1) [43]:(1)ACT=1−OD600nm, t5hOD600nm, t0h×100

### 2.5. Abiotic Support and Microscopic Observation of the Resulting Microbial Organization

#### 2.5.1. Microtiter Plate for Adhesion and Biofilm Formation Capacity

In order to avoid any misleading results that may occur when using microplates treated to enhance cell adhesion, untreated, pyrogenic and polystyrene 6-wells microplates (Sarstedt, Nümbrecht, Germany) were used to analyze adhesion and biofilm formation. For all experiments, microplates were inoculated with 10^8^ CFU/mL of a bacterial suspension in MH broth.

Concerning cell adhesion-related experiments, the medium was removed after 48 h of incubation at 37 °C under anaerobic conditions and the wells were carefully washed using the steam-based method as described by Tasse et al. [44]. Steam was used as a soft washing technique to preserve adherent bacteria and biofilm integrity as well as to improve the reproducibility in quantifying adhesion and biofilm formation capacity. This washing system was operated for 40 min in order to remove the medium and all non-adherent bacteria. The adherent bacteria were resuspended in PBS solution by scraping the well surfaces. The DNA was then extracted and a specific region of the 16S rRNA of *P. distasonis* was amplified and quantified by qPCR (see Section 2.2 for primers and protocol). Knowing the number of 16S operons in *P. distasonis* and controls, we then determined the number of bacterial cells/cm².

For biofilm-related experiments MH was gently removed after 48 h of incubation at 37 °C under anaerobic conditions and the wells were then cleaned with PBS and subsequently reflowed with Schaedler broth, to promote the growth of the adherent bacteria. The microplates were further incubated using the same conditions for additional 6 days. The medium was renewed every 2 days. After 8 days of incubation in total, the biofilm was quantify as described for the adhesion-related experiments. The analysis of adhesion as well as that of biofilm formation were each carried out in triplicate in two independent experiments.

#### 2.5.2. Microscopic Observation

Untreated, pyrogenic and polystyrene 6-wells microplates were inoculated as previously described for biofilm experiments in order to study the organization of *P. distasonis* biofilms. The plates were steam-washed and pyrogenic sterile water was added to the wells. For staining, a working solution of 100× diluted SYBR^®^ Green I (SG) (Thermo Fisher Scientific, Villebon-sur-Yvette, France) was used. 4 mL of the sample was stained with SG working solution at 10 µL·mL^−1^ for 15 min while protecting the plates from light. To remove the excess of SG, a second steam-wash was realized and an aluminum foil was used to protect plates from light. Stained bacteria were recovered in 2 mL of pyrogenic sterile water in order to perform in situ observations of the biofilms under an epifluorescence microscope (Olympus BX51, Life Science Europa GmbH, city, Germany), equipped with a water immersion objective (×600 magnification). A 470–490 nm (blue) excitation filter was used, coupled with a barrier filter at 520–530 nm.

### 2.6. Impact of Stress-Induced Molecules on Bacterial Adhesion and Biofilm Formation

Effects of stress-induced molecules on adhesion and biofilm formation were investigated (in triplicate) using untreated, pyrogenic and polystyrene 96-wells microplates (Sarstedt). Before use, epinephrine, norepinephrine, dopamine and serotonin (Sigma-Aldrich) were dissolved in distilled water to a final concentration of 10 mM. Cortisol (Sigma-Aldrich) was dissolved in distilled water to a final concentration of 250 µM. A 0.2 µm filter was used to sterilize the solutions. The compounds were then used at the following concentrations: epinephrine: 100 µM, norepinephrine: 100 µM, dopamine: 100 µM, serotonin: 100 µM, cortisol: 0.250 µM and were added at two different times: either during the pre-culture phase as described in Section 2.1, or during the inoculation of bacterial cells in plates, corresponding to the beginning of the adhesion/biofilm formation phase.

For these experiments, bacterial suspensions (10^8^ CFU/mL) were inoculated into microplates in adapted medium as previously described. The plates were incubated for 48 h at 37 °C under anaerobic conditions for both adhesion and biofilm formation assay. Using the steam-based method, the wells were then washed as previously described. Plates were dried for 10 min at 60 °C and stained for 5 min with 150 µL of 2% Gram’s crystal violet solution (Sigma-Aldrich) like previously described by Donelli, et al. [45]. The excess stain was then rinsed for 40 min using steam-wash. After 10 min drying at 37 °C, the adherent bacteria were resuspended in 150 µL of 33% (*v/v*) glacial acetic acid (Merck, Darmstadt, Germany) and analyzed by OD_570nm_ measurement.

### 2.7. Electrophoretic Mobility Measurements

Electrophoretic mobility experiments consist in following the displacements of particles in a quartz Suprasil^®^ rectangular capillary (Helma, Jena, Germany) upon application of a constant direct-current electric field (800 V/m). Particle scattering tracking is monitored by reflection of a laser beam at 90° angle via a charge-coupled device (CCD) camera, with trajectories recorded in real time and processed by CAD image analysis software to derive electrophoretic mobility distributions [46].

Assays were incubated for 16 h on Schaedler broth, washed by centrifugation and resuspended in 1 mM NaNO_3_. The washed bacterial suspensions were then diluted and added in 1 mM to 250 mM NaNO_3_ in order to have a bacterial cell number of 10^2^ in the quartz electrophoresis cell.

Electrophoretic mobility measurements were performed at natural pH and room temperature in triplicate in two independent experiments for each NaNO_3_ electrolyte concentration tested using a Zetaphoremeter IV (CAD Instruments, Les Essarts-le-Roi, France). The electrophoretic mobility *μ* (m^2^·V^−1^·s^−1^) of bacterial cells may be approximated at sufficiently large electrolyte concentrations in line with the establishment of Donnan electrostatics in the electrokinetically active peripheral cell surface layer, by the following expression [47]:(2)$$\mu =\frac{\rho_{o}}{\eta \lambda_{o}^{2}}+\frac{\varepsilon }{\eta }\frac{\psi^{o}/\kappa_{m}+\psi^{D}/\lambda_{o}}{1/\kappa_{m}+1/\lambda_{o}}$$
where $\rho_{o}$(C × m^−3^) represents the effective density of charges carried by the cell-layer, $\kappa_{m}$ (m^−1^) the reciprocal Debye layer thickness in that layer and $\lambda_{o}$ the softness parameter. ε (F·m^−1^) and η (Pa·s^−1^) refer to the dielectric permittivity and viscosity of the medium (with ε = 8.854 × 10^−12^ F·m^−1^ and η = 0.96 × 10^−3^ Pa·s^−1^ under the temperature conditions of interest in the work). The quantity $1/\lambda_{o}$ (m) corresponds to the characteristic penetration length of the electroosmotic flow developed under electrophoresis conditions within the soft permeable surface structure. In Equation (2), $\psi^{o}$ (V) corresponds to the surface potential, i.e., the potential at the position corresponding to the location of the outer boundary of the surface layer, and $\psi^{D}$ (V) is the Donnan potential, i.e., the electrostatic potential reached within the bulk of that layer. The parameters $\psi^{D}$, $\psi^{o}$ and $\kappa_{m}$ all depend on the space charge density $\rho_{o}$ according to the following expressions:(3)$$\psi^{D}=\frac{RT}{F}{\sin h}^{-1}\left(\frac{\rho_{o}}{2FI}\right)$$
(4)$$\psi^{o}=\psi^{D}-\frac{RT}{F}\tan h\left(\frac{F\psi^{D}}{2RT}\right)$$
(5)$$\kappa_{m}=\kappa {\left\{\cos h\left(\frac{F\psi^{D}}{RT}\right)\right\}}^{1/2}$$
where κ represents the reciprocal of the screening Debye layer thickness, *R* the gas constant, *T* the absolute temperature, *F* the Faraday number (C.mol^−1^) and *I* the solution ionic strength fixed in our experiments by the NaNO_3_ electrolyte concentration. $\rho_{o}$ and $1/\lambda_{o}$ were determined from the measured variation of *μ* with changing *I* using standard Levenberg-Marquardt procedure for fitting cell electrophoretic mobility data to Equations (2)–(5).

### 2.8. Statistical Analysis

The one-way analysis of variance (ANOVA) was used to group the samples according to their adhesion or biofilm formation capacities. In order to determine whether stress-induced molecules significantly impact adhesion and biofilm formation, a *t*-test was adopted. The *p* values < 0.05 were considered as statistically significant. The principal component analysis (PCA) was used to highlight the potential correlation between the different variables studied and the hierarchical agglomerative clustering (HAC) to cluster *P. distasonis* strains based on the similarity of their capacities. All statistical analyses were carried out with XLSTATs program version 2021.2 (Addinsoft, Paris, France).

## 3. Results

### 3.1. Survival Capacity of P. distasonis during Stressing Conditions Linked to Experiments: Aerobic Condition, Saline Solution, Diluted MH and NaNO_3_ electrolyte

In the experiments reported in this study, the protocols included cell manipulation in aerobic and other potentially stressful environments (physiological water, diluted MH and NaNO_3_ electrolyte). To address whether these conditions may cause bacterial death, the ability of the cells of interest to survive under such conditions was investigated (Appendix A).

Cell enumeration assays performed after 2, 4, 6, 24 and 48 h under aerobic condition and at room temperature in saline solution showed that *P. distasonis* DSM 20701^T^ was able to survive under such conditions at least for 6 h without any indication of mortality. This coincides with the presence of putative cytochrome D ubiquinol oxidase subunit I and II genes annotated on the genome of *P. distasonis* DSM 20701^T^ (BDI_2648; BDI_2649).

We also observed that, under anaerobic condition, *P. distasonis* was able to survive 48 h in a diluted MH medium without any growth. In the same way, NaNO_3_ concentration conditions adopted for electrokinetic measurements did not impact on the bacterial growth during the 1-h duration of the experiments.

### 3.2. Homo-Aggregation Capacity of P. Distasonis

*P. distasonis* strains exhibited homo-aggregation capacities ranging from 4.6% to 8.2% after 5 h which were significantly lower than that observed for the positive control LGG (14.7%) (Figure 1A).

### 3.3. Adhesion and Biofilm Formation Capacities of P. distasonis Strains on Abiotic Support

Two significantly different groups could be identified from the adhesion assays (Figure 1B): the first one, denoted as (a), with the CS1 and the positive control LGG, and the second one, denoted as (b), that includes thirteen out of the fourteen *P. distasonis* strains. In this study, all tested strains showed adhesion capacities to an abiotic support after 48 h of incubation. The adhesion capacity of the 10^8^ inoculated bacteria/cm² ranged from 7 × 10^2^ adherent bacteria/cm² (strain CS12) to 7 × 10^5^ adherent bacteria/cm² (with the strain CS1 showing an adhesion capacity as strong as that of the positive control LGG).

Concerning biofilm formation, the experiments showed that although all tested strains had the capacity to form a biofilm, an inter-strain variability was observed (Figure 1C), similarly to results pertaining to the adhesion capacity. The statistical comparison revealed two significantly different groups ranging from 2 × 10^5^ to 4 × 10^8^ bacteria/cm². The first group (a) included the CS8 while the second group (b) included thirteen additional *P. distasonis* strains. The positive control *Bf* 638R belonged to both groups with a bacterial cell number of approximately 3 × 10^8^ bacteria/cm². The negative control *Bt* VPI-5482 did not form any biofilm after 48 h of incubation, its DNA concentration being below the detection limit (10^3^ bacteria/mL) for the qPCR.

Additionally, although all *P. distasonis* strains tested formed biofilm, epifluorescence microscopic observations highlighted some differences in terms of biofilm organization (Figure 1D). Two distinct types of biofilm organization, different from the control strain *Bf* 638R, were observed for all the tested strains (Figure 1C): the first one included well-defined patches of single cells and of small clusters (a), and the second one showed larger bacterial clusters (b). *Bf* 638R biofilm was covering the entire surface of the well (c) whereas *Bt* VPI-5482 showed only a few single cells distributed over the entire abiotic surface (d). Overall, experiments on *P. distasonis* capacities revealed an important inter-strain variability for both adhesion and biofilm formation properties and biofilm structure.

### 3.4. Electrokinetic Properties of P. distasonis

The electrokinetic cell surface properties were estimated for the fourteen *P. distasonis* strains and the three control strains used in this study from proper fitting (Equations (2)–(5)) of the dependence of their electrophoretic mobility on NaNO_3_ electrolyte concentration. For the sake of illustration, results are shown in Figure 2 for selected bacterial strains while profiles for all tested strains are reported in Appendix A. The theoretical electrophoretic mobility curves deviated from experiments below a threshold value of electrolyte concentration (ca. 10–20 mM). In addition, electrophoretic mobility tended asymptotically to a non-zero plateau value at sufficiently large electrolyte concentrations. Based on the confrontation between theory and experiments (Table 1), the charge density for CS1 was remarkably lower (−8.6 mM) than the charge density of the other *P. distasonis* strains tested (between −33.8 and −23.7 mM). The charge density of CS1 was further close to that of LGG (−7.0 mM), and, in less extent, of the biofilm formation positive control strain *Bf* 638R (−12.5 mM) while other strains had charge densities with order of magnitude comparable to *Bt* VPI-5482 (−27.3 mM). Based on the hydrodynamic penetration length scale 1/λ_o_, a significantly higher surface permeability was observed for the CS1 (1/λ_o_ of 2.51 nm) compared to that of all other strains examined. The surface permeability of CS6 was also quite high with a 1/λ_o_ of 2.23 nm. All other *P. distasonis* strains exhibited a surface permeability ranging between 1.60 nm (CS2) and 1.98 nm (CS8). The 1/λ_o_ of the control strains *Bf* 638R and *Bt* VPI-5482 were in the same range (1.91 and 1.86 nm, respectively). Then, the hydrodynamic penetration length of LGG is the smallest evaluated with 1.07 nm.

### 3.5. Correlation between ACT, Electrokinetic Surface, Adhesion and Biofilm Formation Properties of P. distasonis

PCA permitted us to analyze the relationship between homo-aggregation, adhesion, biofilm formation and electrokinetic properties of the fourteen *P. distasonis* strains tested (Figure 3). The 2-D PCA explained a total of 77.35% of the total variations in the dataset. The positive and negative controls (LGG, *Bf* 638R and *Bt* VPI-5482) used in this study were left out of the PCA analysis to avoid a masking effect caused by their significant differences from *P. distasonis*.

The 2-D PCA (Figure 3A) suggested a positive correlation between the charge density of the cell surface and the adhesion capacity on abiotic support with a narrow angle close to 0°. Indeed, CS1, that displayed the highest adhesion capacity, had also a significantly lower density of charges (and a larger cell surface permeability), suggesting the implication of rather extended and loose surface structures in its adhesion to abiotic substrate. A negative correlation between the ACT and the cell surface permeability may appeared as the CS1 and CS6 with the highest surface permeability also displayed the lowest ACT. However, as the difference between the ACTs of CS1, CS6 and the other strains were not significant, this correlation cannot be considered generic. The right-angles (=90°) between other parameters indicated the independence and the absence of relationship between the variables. With the exception of CS1, CS6 and CS8, results were relatively similar. The clustering of *P. distasonis* strains by HAC (Figure 3B) conformed these observations by revealing five distinct clusters where the CS1, 6 and 8 were individually grouped. Despite differences in electrokinetic properties among all strains, the ACT of CS1 was not different from that of other *P. distasonis* strains. In the same way, the CS13 and the CS6, which exhibited respectively the highest and lowest ACTs, did not show any difference in their adhesion or biofilm formation capacities, demonstrating the absence of correlation between ACT and other surface properties. The CS7 and the CS8, which had the highest biofilm formation capacities, had ACTs and electrokinetic properties comparable to those evaluated for the other strains. Similarly, CS1 which had the lowest density of charge and the highest surface permeability, and CS13 which had the highest ACT did not show particularly strong biofilm formation capacity.

### 3.6. Impact of Stress-Induced Molecules on Adhesion and Biofilm Formation

The impacts of the main stress-induced hormones and neurotransmitters released in the gut lumen on the adhesion and biofilm formation capacities of *P. distasonis* were investigated, in order to determine whether they may interfere with the persistence of *P. distasonis* within the GM. We observed that all the tested molecules had an effect, either positive or negative, on the adhesion and biofilm formation capacities of several *P. distasonis* strains (Figure 4). All but the type strain showed at least one response (positive or negative) to the stress markers, either on the adhesion (35% of the assays) or the biofilm formation (23% of the assays). Interestingly, results for most strains also depended on the time when molecules were added (i.e., during growth phase, adhesion phase or during biofilm formation). Basically, a significant reduction in the adhesion and the biofilm formation capacities was observed in respectively 29% and 14% of the assays, while respectively 6% and 9% showed a significant increase in these capacities. Reduction in the adhesion capacity occurred primarily when the molecules were added during the growth phase (19% *versus* 10% of reduction observed when molecules were added during the adhesion) whereas the reduction in biofilm formation occurred primarily when molecules were added during biofilm formation (10% *versus* 4% of reduction observed when molecules were added during the growth phase). Unexpectedly, a stress-induced effect on the adhesion capacity was rarely correlated with the same effect on the biofilm formation capacity.

## 4. Discussion

*P. distasonis* is a member of the gut core microbiome that brings many beneficial properties for its host, on the metabolic, immune and intestinal inflammation aspect [15,18,19,20,48]. Its relative abundance within the GM has however been negatively associated with various chronic inflammatory diseases [48,49,50,51]. Interestingly, its abundance was also significantly decreased in chronic stress situations, being involved in the onset or exacerbation of such chronic disorders [8,10,11]. However, little is known so far about the properties of *P. distasonis* that could explain its persistence or loss within the GM.

Bacteria must reach a support before they can interact with it. In a given environment, the homo-aggregation coefficient is a measure of the ability of a bacterial organism to form microcolonies with other organisms of the same species. As a given microorganism’s ability to aggregate increases, the more it will sediment and reach the support [52]. Moreover, the ability of cells to form aggregates is a required condition for the formation of multicellular clumps, commonly considered to be one of the first steps in biofilm formation. Cell surface molecules, such as proteins and exopolysaccharides, may also be involved in cell-cell interactions and thereby homo-aggregation. This study demonstrated that all strains of *P. distasonis* investigated were able to homo-aggregate moderately, showing their potential to sediment and consequently, reach and adhere onto a given support.

The second step, after reaching the support, is the adhesion of the bacteria. The adhesion capacity of the bacteria to a support may be due to occurrence of specific interactions involving cell surface molecules and structures (pili, fimbriae, flagella, auto associative proteins, exo/lipopolysaccharides and many others) or unspecific interactions like van der Waals forces, hydrophobic and electrostatic interactions [53,54]. These interactions directly depend on the support composition on which bacteria are located, including its surface charge, its hydrophobicity and its chemical surface functionalities. In our study, untreated, polystyrene-plates were used, showing an uncharged surface with hydrophobic groups which eliminates the potential involvement of surface charges in the adhesion process. Under the tested conditions, all *P. distasonis* strains showed an adhesion capacity, with one of the studied strains revealing an adhesion capacity as important as the positive control *L. rhamnosus* GG.

After the colonization of a surface by planktonic cells, a conversion into a sessile lifestyle is essential for bacteria in order to protect themselves against environmental stress (physical stress, antibiotics, etc.). This protection is provided by a matrix usually composed of exopolysaccharides, that improve the capture of nutrient by bacteria, as helped by the spongy structure of the exopolymeric matrix and by an increased enzyme retention that leads to more-efficient substrate conversion [55]. The bacterial transition from planktonic lifestyle to biofilm organization requires changes in gene expression and physiological modification that ensure the production of the extracellular matrix and the up/down-regulation of genes involved in the synthesis of factors required for biofilm formation, such as adhesion factors needed for the settling of the adhesion step [56]. In this study, we demonstrated that all the *P. distasonis* strains studied had the capacity to form a biofilm. Different organization of the biofilm could be observed among the strains. Not surprisingly, the organization seems to depend on the number of bacteria in the biofilm: the more important is the quantity of bacteria, the larger bacterial clusters will be observed.

The investigation of the electrokinetic properties of the cell peripheral regions (outer membrane and anchored surface structures) may provide a first qualitative picture of the surface determinants driving adhesion and biofilm formation. The electrophoretic mobility of all tested strains was negative and showed a plateau at sufficiently large ionic strengths. This asymptotic plateau value is the signature for the presence of a charged and permeable layer on the bacteria surface [46]. Gram-negative and Gram-positive bacteria show important differences in terms of cell wall structures. Differences between the positive control LGG (Gram-positive) and *P. distasonis* (Gram-negative) strains could thus be expected. Interestingly, the CS1 strain showed an external surface that was less electronegative than the other strains, with a charge density that was close to that of LGG. The electrokinetic features of LGG are due to the presence of exopolysaccharides and pili at its surface, which could suggest the possible presence of such surface structures for the CS1 strain that is defined by similar adhesion capacity [46]. Other *P. distasonis* CS revealed charge densities closer to that of *Bt* VPI-5482. *B. fragilis* 638R, that show an external surface slightly less charged than the majority of *P. distasonis* strains but a close surface permeability, is well-described for its 10 distinct polysaccharide (PS) biosynthesis loci [57]. Previous studies on several strains of *B. fragilis* have further shown the presence of fimbriae, lipopolysaccharides with particular structure, OmpA and outer-membrane vesicles on its surface, all potentially involved in its adhesion [58,59]. In *P. distasonis* DSM 20701^T^, 13 capsular PS biosynthesis loci have been described with four of them that are disrupted by phage insertions between the upstream regulatory genes (*upcY* and *upcZ*) and the downstream PS biosynthesis gene cluster of capsular PS [15]. No study has yet highlighted the presence of other external structures involved in the adhesion and biofilm formation capacity of *P. distasonis*. Just one study revealed the absence of fimbriae-like structure in *P. distasonis* as well as *B. fragilis* [60].

The expression of surface components of *B. fragilis* is subject to phase variations, a reversible ON-OFF phenotype, controlled by DNA inversions of the polysaccharide biosynthesis loci promoters. These abilities allow the bacteria to elaborate an extremely flexible and adaptive surface architecture that is likely pivotal for their long-term survival and predominance in the human colon [57,61,62]. In *P. distasonis*, such a mechanism has been described for a phase-variable S-layer glycoprotein and for 11 of the 13 capsular PS biosynthesis loci, with a tyrosine-type site-specific recombinase encoding gene upstream the *UpxY* homolog gene [15,62]. The same study revealed an important proportion of laterally transferred genes in *P. distasonis* genome, potentially involved in its maintenance within the GM [15].

Our findings give a first qualitative insight into the adhesion and biofilm formation capacities of a large panel of *P. distasonis* strains and provide some hypothesis about the mechanisms that could explain its persistence in the digestive tract. However, the experiments performed on an abiotic support, obviously, cannot reflect on their own the complexity of the digestive tract as the pH effects and/or presence of adhesins at the surface of epithelial cells may considerably affect (positively or negatively) the adhesion and biofilm formation capacities. Bacterial surface structures are decisive to mediate interactions with other microorganisms as well as with the host’s cells [40,52]. Except the correlation between the charge density of the cell surface and the adhesion capacity suggesting the implication of rather extended and loose surface structures in the CS1 adhesion capacities to abiotic substrate, no clear and generic correlation between the other cell properties has been found. This is an important result as such. Indeed, our observations can be explained by the very diversity in the structure of *P. distasonis* surface that is inherently related to the inter-strain variability observed in all our experiments. This underscores a large panel of cell surface arsenal and corresponding flexible cell adhesion capacities: the very diversity of *P. distasonis* could be a decisive property that promotes its persistence in gut via efficient adhesion and subsequent biofilms formation abilities. However, it is important to specify that the study relates to the adhesion and to the early biofilm formation phases of *P. distasonis*. More precise analysis of the biofilm architecture is required for evaluation of the biofilm development and the role of surface structures in cell adhesion [63,64].

As we previously described in murine models, a chronic stress led to a decrease in intestinal relative abundance of *P. distasonis* [10,11]. This reduction may be due to the action of stress markers released in the intestine that have been described to modulate bacterial adhesion and biofilm formation [31,32,53,65]. In the present study, we have investigated the potential effect of such markers on the adhesion and biofilm formation capacities of *P. distasonis.* Our results showed that all the molecules tested, hormones and/or neurotransmitters, can either reduce or improve its adhesion and biofilm formation capacities, depending on the strain and the molecule tested. These findings suggest the presence of sensing elements at the bacterial surface that are potentially activated/triggered by host signal molecules. The sensor kinase QseC, part of the two-component regulator system in enterohemorrhagic *Escherichia coli,* is well-described to act as an adrenergic receptor for host catecholamines [66,67]. Signal-sensing systems have also been described in anaerobic bacteria with, e.g., the LuxS/Autoinducer-2 quorum sensing system in *Porphyromonas gingivalis* or the LuxR homologues in *B. fragilis* [68,69]. However, although an impact of catecholamines has already been observed in *P. gingivalis* [70,71], no study has yet demonstrated the potential implication of these signal-sensing systems in the response of anaerobic bacteria to host stress molecules. In the public sequenced genomes of *P. distasonis* (Microbial Genome Annotation and Analysis Platform, LABGeM, CEA), only one putative LuxR with 39–40% synteny with the *B. fragilis* LuxR was found. The presence of such signal-sensing system in *P. distasonis* could explain the potential impact of the stress-induced molecules on its persistence through the GM.

## 5. Conclusions

Although there is growing evidence of the beneficial roles of *P. distasonis* within the GM, little is known about its capacities to interact with other bacteria, to adhere to surfaces and to form biofilms. This study, performed on a wide diversity of strains, allows a better knowledge of the homo-aggregation, adhesion and biofilm formation capacities of *P. distasonis* and potential mechanisms involved in it. Under the conditions of interest in our work, all tested strains showed capacities to homo-aggregate, to adhere and to form a biofilm with an important inter-strain variability. These results, in addition to the high adaptability of *P. distasonis* demonstrated in previous studies, give a first explanation to its persistence within the GM. The evaluation of the impact of five stress-markers on the adhesion and biofilm formation capacities of *P. distasonis* also revealed a remarkable strain variability with either positive or negative effects, depending on the molecule tested. Interestingly, catecholamines mainly reduced in vitro adhesion capacity of *P. distasonis* on abiotic support, which coincides with the in vivo stress-induced abundance decrease that we previously described in murine models. Further researches on the nature of the external structures harbored by the various *P. distasonis* strains and the existence of potential signal-sensing systems of this species could be valuable for a better understanding of the molecular pathways explaining how a chronic stress impact *P. distasonis*. Such data will further help in designing ways to counteract the stress-induced loss of this beneficial species within the digestive tract and to promote its use in next-generation probiotics.

## Figures and Tables

**Figure 1 microorganisms-09-01602-f001:**
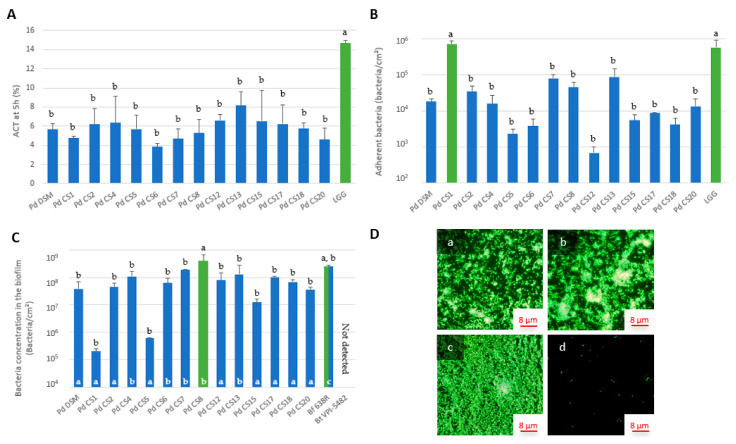
(**A**) Homo-aggregation coefficient (ACT) of 14 strains of *P. distasonis* and LGG (positive control) after 5 h of incubation at 37 °C under anaerobic conditions. The colors coupled with a and b indicate the significantly different groups after ANOVA, *p* < 0.05. Error bars reflect standard error of the mean. (**B**) Adhesion capacity of 14 strains of *P. distasonis* and LGG (positive control) on abiotic support after 48h of incubation at 37 °C under anaerobic conditions. The colors coupled with a and b indicate the significantly different groups after ANOVA, *p* < 0.05. Error bars reflect standard error of the mean. (**C**) Biofilm formation capacity of 14 strains of *P. distasonis*, *Bf* 638R (positive control) and Bt VPI-5482 (negative control) on abiotic support after 6 days of incubation at 37 °C under anaerobic conditions. The colors coupled with black a and b indicate the significantly different groups based on ANOVA, *p* < 0.05. White a, b, and c indicate the biofilm organization pictures collected in (**D**). Error bars reflect standard error of the mean. (**D**) Biofilm organization on abiotic support after 6 days. (**a**) Organization showing small clusters of bacteria from *P. distasonis* CS2 biofilm; (**b**) Organization showing large clusters of bacteria from *P. distasonis* CS7 biofilm; (**c**) Biofilm formed by the positive control *Bf* 638R; (**d**) Scattered cells from the negative control *Bt* VPI-5482. All the data represent mean ± standard error or observations from at least two independent experiments performed with each strain in triplicate (n = 2 × 3).

**Figure 2 microorganisms-09-01602-f002:**
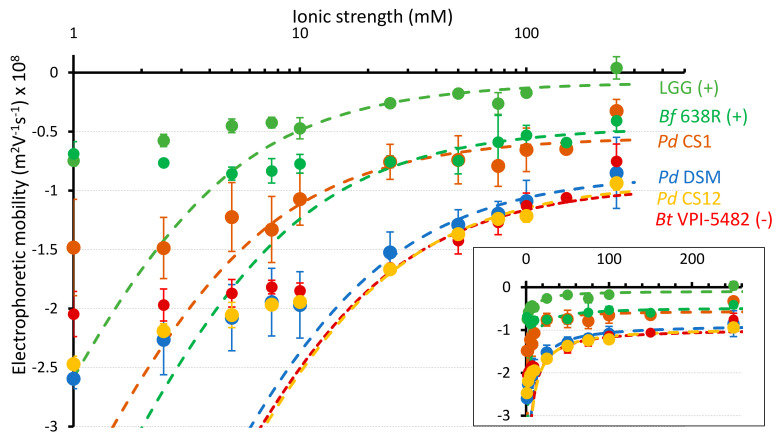
Impact of electrolyte concentration on the electrophoretic mobility of 3 *P. distasonis* strains (CS1, CS12 and DSM 20701^T^) and controls LGG, *Bf* 638R and *Bt* VPI-5482. Points: experimental data. Dotted lines: fitting of the experimental electrokinetic data by the theoretical expressions 2-5 for the electrophoretic mobility of soft particles. In inset, electrokinetic data are represented according to linear axis in electrolyte concentration. Error bars reflect standard error of the mean. Data represent mean ± standard error from at least two independent experiments performed with each strain in triplicate (n = 2 × 3).

**Figure 3 microorganisms-09-01602-f003:**
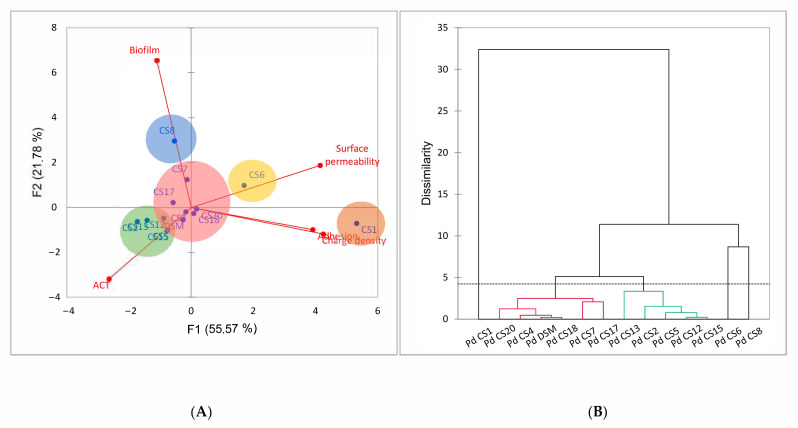
(**A**) Two-dimensional principal component (2-D PCA) analysis biplot showing relationships amongst the charge density ($\rho_{0}$/F), the surface permeability (1/*λ*_o_), the homo-aggregation coefficient (ACT), the adhesion and biofilm formation capacities of *P. distasonis* strains with a F1 and F2 of 77.35%. (**B**) Dendrogram representing clustering of *P. distasonis* strains based on the similarity of their capacities analyzed by hierarchical agglomerative clustering (HAC). The dotted line characterizes the automatic truncation, leading to five groups identified with circles in the 2-D PCA.

**Figure 4 microorganisms-09-01602-f004:**
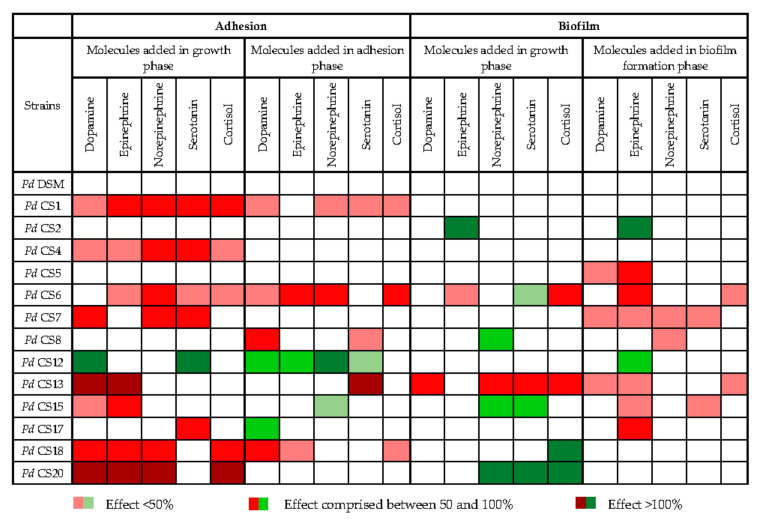
Impact of dopamine, epinephrine, norepinephrine, serotonin and cortisol on the adhesion and biofilm formation capacity of 14 strains of *P. distasonis.* Pd: *P. distasonis*. Red square indicates a significant reduction of the adhesion/biofilm formation capacities of the strains. Green square indicates a significant improvement of the adhesion/biofilm formation capacities of the strains. White square indicates no impact of the stress-induced molecules on the properties of the strains. Statistical analyses were done using the *t*-test. Data represent mean from triplicate for each condition (n = 3).

**Table 1 microorganisms-09-01602-t001:** Charge density ($\rho_{o}$/*F*, expressed in concentration of equivalents (anionic) charges) and cell surface permeability (1/λ_o_) of *P. distasonis* and control strains.

Strains	$\rho_{0}/F\;(\mathrm{mM})$	1/*λ*_o_ (nm)
*P. distasonis* DSM 20701^T^	−26.6	1.79
*P. distasonis* CS1	−8.6	2.51
*P. distasonis* CS2	−37.0	1.60
*P. distasonis* CS4	−30.4	1.80
*P. distasonis* CS5	−23.5	1.85
*P. distasonis* CS6	−23.7	2.23
*P. distasonis* CS7	−29.0	1.84
*P. distasonis* CS8	−32.3	1.98
*P. distasonis* CS12	−29.1	1.78
*P. distasonis* CS13	−33.8	1.80
*P. distasonis* CS15	−29.1	1.78
*P. distasonis* CS17	−33.8	2.04
*P. distasonis* CS18	−24.0	1.88
*P. distasonis* CS20	−27.4	1.87
*L. rhamnosus* GG ATCC 53103	−7.0	1.07
*B. fragilis* 638R	−12.5	1.91
*B. thetaiotaomicron* VPI-5482	−27.3	1.86

## Data Availability

Data is contained within the article or Appendix A.

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
