# Peer review of "Surface Properties of Parabacteroides distasonis and Impacts of Stress-Induced Molecules on Its Surface Adhesion and Biofilm Formation Capacities"

_microorganisms, 2021, doi:10.3390/microorganisms9081602_

Round 1

Reviewer 1 Report

REVIEW OF THE ARTICLE BY JORDAN CHAMARANDE ET AL. ENTITLED ‘SURFACE PROPERTIES OF PARABACTEROIDES DISTASONIS AND IMPACTS OF STRESS-INDUCED MOLECULES ON ITS SURFACE ADHESION AND BIOFILM FORMATION CAPACITIES’ (microorganisms-1287789)

Jordan Chamarande et al. evaluated adhesion of the cells of the strains of gut bacterium Parabacteroides distasonis as function of electrokinetic properties of cells. They studied adhesion of the cells to abiotic surfaces (polystyrene and pyrogenic plates) and discuss biofilm-forming potential. To describe their results, the Authors used classical electrokinetic theory. They calculated fundamental electrostatic parameters of their bacterial cells (charge density and surface permeability). They also followed the addition of ‘stress-related molecules’, such as serotonin and dopamine, on adhesion characteristics. Physicochemical interactions, such as electrostatic interactions, play a very important role in primary stages of cell adhesion and possible further biofilm formation. Knowledge about them is important for understanding of bacterial surface-to-surface interaction. Therefore, obtained data are valuable. The study of physicochemical properties of bacterial cells is an interesting poorly studied issue. The article is in scope of the journal. At the same time there are some drawbacks. Thus, I suggest major revision of the text in accordance with my comments (see below). My main criticisms are addressed to experimental evidence of true biofilm formation.

GENERAL COMMENTS

-Actually, there is no experimental evidence of ‘true’ biofilm formation. No biofilm formation observed. EPS detection is required to show it. Thus it might be simple collective adhesion of cells to a surface. Please, explain and discuss. In addition, 

-Unfortunately, I could not see any supplementary materials for the fork.

-’homo-aggregation’ - do You mean self-aggregation? Please explain importance of this phenomenon for biofilm development (see e.g. Costerton (1995) Annu Rev Microbiol 49(1):711–745; Nikolaev, Plakunov (2007) Microbiology, 76(2), 125-138; Sriramulu et al. (2005) Journal of medical microbiology, 54(7), 667-676). 

-When describing the number of bacteria, please use the term cell number instead of concentration, because concentration refers to homogeneous solutions. 

-Avoid references in results.

INTRODUCTION

-l. 38-39. Please, clarify, that it is the data for the human gut.

-l. 38-39,53. Bacterial taxa higher than family should not be italicized.

-l. 39-41. It is not the only function of gut bacteria. They virtually play the role in functioning of virtually all systems in the organisms and in many processes. See e.g. Egerton et al. (2018) Frontiers in microbiology, 9, 873.

-l. 46. Please, change '…' to a more scientific 'and many others'.

-l. 54. ‘core microbiomes’ should not be italicized.

-l. 64-74. Here it is better to introduce a reader to the main concepts of biofilm theory. It would be especially nice to provide definitions of biofilm, EPS, its main components. Main stages of biofilm formation (cell adhesion/co-aggregation, microcolony formation, formation of matured biofilm, biofilm dissociation and propagation). Please, add most common references in the field, e.g.  Costerton (1999) Int. J. Antimicrob. Agents 11, 217–221 (your reference [24]) or more detailed review (Costerton (1995) Annu Rev Microbiol 49(1):711–745; Nikolaev, Plakunov (2007) Microbiology, 76(2), 125-138.).

-l. 65-68. Here it is important to define the concepts of reversible and irreversible adhesion and describe the role of 'non-specific' physicochemical forces and 'specific' biochemical processes (with corresponding references).

-Please, clarify in more detail, why is it too important to study precisely Parabacteroides distasonis. For example, the first paragraph of Discussion is more suitable for Introduction.

-Please, unify ‘electrokinetic’ and ‘electrodynamic’. ‘Electrokinetic’ is more appropriate.

MATERIALS AND METHODS

-l. 91. 15% of what? Of culture volume?

-l. 103. Please, indicate the country of origin for DNA MiniKit.

-l. 110-111. Plese, check Your amplification profile. Is 40 cycles reffered to “95 °C for 15 seconds, 60 °C for 1 minute” but not 95 °C 5 min? Where is the elongation phase?

-l. 114. %- wt/v?

-l. 114-115. Please, provide the reference for “Mueller-Hinton (MH) (1/10) and NaNO3 electrolyte”

-l. 114. 1/10 - is it dilution - please, specify.

-l. 114-115 “NaNO3 electrolyte” - what does it mean? Plese, indicate at least Your concentration.

-l. 127-134. It is better to move this information to the Introduction to define Your terms for a reader (see my general comments).

-l. 136. Indicate, how did You determine optical density? In a spectrophotometer? In standard quartz cuvettes?

-l. 139. The expression is not numbered. Why is it in bold?

-l. 145. Was should be were.

-l. 146. ‘, microplates plates ’ - please, revise.

-l. 168. ‘final concentration of 1X’ - what does it mean? Please, indicate the final concentration of Sybr Green in Your samples.

-l. 173. Please, indicate Your light source for fluorescence excitation and the spectral channel for fluorescence detection. Did You use light filters?

-l. 173. Indicate manufacturer and country of origin for the Olympus BX51 microscope.

-l. 177. Please, indicate manufacturer and country of origin for e 96-wells microplates.

-l. 177-180. Please, justify here the selection of ‘stress-related’ molecules.

-l. 181. It is better to say ‘the compounds’ instead of ‘The molecules’.

-l. 196-198. Indicate the parameters of Your electric field.

-l. 197. Quasi stationary electric field?

-l. 199. ‘incubated for 16 hours on Schaedler’ - it is a jargonism, please, revise. Schaedler agar?

-l. 200. ‘NaNO3 1mM’ should be ‘1mM NaNO3’.

-l. 205. Please, indicate country of origin.

-l. 205. ‘According to theory,’ - remove.

-l. 209-218. In eq. 1 You also should explain ε and η. 

-l. 209. What is ε? Dielectric constant? If yes, why did you not take into account doelectric permeability of the medium?

-l. 219. I was confused by units in Eq. 2. What were the units of the ion strength? Commonly, they are calculated as mol/g, however it will give l/g in sinh-1. Please, explain.

-l. 221. In Eq. 4 κ also should be explained.

-l. 209-226. For comprehension, it is better to give units for all parameters.

-. 227-234. The procedure of clusterization also should be added.

-Please, describe in more detail the procedure of adhesion capacity determination. Did You calculate the cells? How? The same is true about cell number in the biofilms.

RESULTS

-l. 249. ‘observe’ should be ‘observed’.

-l. 255-257. It is not Results. Please, move to either Introduction or Discussion.

-Figurte 2. Replace ‘E’ to powers of 10.

-l. 284.  3.108 bacteria/cm² where? In the biofilm?

-Figure 3: figure caption cannot contain a reference to another figure.

-Figure 4: please indicate the clusters (l. 293-300) on the figure. 

-Figure 4: Figure panels should be named by letters (not numbers).

-You can highlight and describe in more detail specific structures of biofilms (pores and voids) on Your figure as a sign of a biofilm (see, e.g. Burtseva, et al.. (2021) Microbial Ecology, 81(4), 932-940).

-l. 298. ‘one large cluster’ - it sounds strange. Do You mean completely developed biofilm covered entire surface of the well?

-l. 319-324. It is discussion. Moreover, the last statement is without a reference.

-Figure 5. ‘Dotted lines: theory’ replace to ‘Dotted lines: fitting by the expressions for ...’.

-Figure 5. What is the value of the insertion on the figure?

Table 1. Please, write bacterial names in full.

Table 1. What was the accuracy of charge density determination? Why there are different numbers significant figuress? Do You actually mean e.g. -7.00 mM instead of -7 mM?

-l. 354. ‘cell surface and the adhesion capacity’ - it is important to clarify, on which abiotic surface.

-Figure 6b. Please, indicate what the numbers on the Y-axis mean?

-l. 398. ‘Surprisingly,’ - it does not sound scientific. Please, remove.

-Table 2: actually, it is not a table, because it does not contain numbers. It is a figure (heatmap). Moreover, it does not correspond to the journal's format of tables.

DISCUSSION

-l. 408. ‘core microbiome’ should not be italicized.

  1. 413. ‘P. distasonis’s’ should be ‘P. distasonis

-l. 422-423, 429-430. proteins and exopolysaccharides are not surface structures. They are molecules, one of the main components of biofilm EPS. Surface cell structures of bacteria are pilli, fimbriae, curli and flagella.

-427-437. It is very important here to briefly describe the chemical nature of the abiotic surface and possible functional groups of it and of bacteria which might be involved in the electrostatic interactions.

-l. 430, 440. Please, change '…' to a more scientific 'and many others'.

-l. 473-476. Flagella also may be involved in adhesion, see Hobley  et al.  (2015) FEMS Microbiol Rev 39(5):649–669.

  1. 487-503. References are missing in the paragraph.

-l. 487-503. Please, highlight, that You focus only on primary stages of cell adhesion, whereas more precise analysis of the biofilm architecture is required for evaluation of biofilm development, as well as the role of surface structures in cell adhesion and building up of cell charge, see and discuss Burtseva, et al.. (2021) Microbial Ecology, 81(4), 932-940; Konduri et al. (2021) Microorganisms, 9(6), 1124.

- l. 487-503. Please, discuss also other physicochemical interactions in the building up of the surface-to-surface contact of P. distasonis (Wan der Vaaltz forces, hydrophobic interactions): Busscher et al. (1997) Advances in dental research, 11(1), 24-32.

Author Response

We sincerely thank Reviewer#1 for her/his positive and relevant comments.

Reviewer 2 Report

An overall well written manuscript with appropriate background and citations. However, the style of the introduction/discussion are very different compared to the results. The latter appear very stern and blunt and the reader would benefit from more background, reasoning of the experiments and short conclusions for the story line.

I suggest some minor changes in the data presentation:

Figures:

Figure 1-4 (maybe 5) could be arranged in a single panel separated by letters. This would allow for easy comparison of the strains between the different measures.

The shades of blue between the different groups in Fig. 1-3 are hardly recognizable. A different color would be appreciated.

Neither the Figures, captions nor the text stated how many replicates were used for any of the experiments making it hard to follow the trends. Especially the data in Fig. 2 + 3 suggest more than just 2 significantly different groups (e.g. Pd CS5). Maybe these differences would be highlighted if the authors normalized their data to the positive control as a percent adhesion (in case of Fig. 2), while reporting the raw values in a supplementary table.

Figure 4: How were the different types characterized other than subjective measure? Number of cells/cluster? Cluster size in um? How many replicates were used for the determination?

Figure 5: In order to emphasize the differences in electrophoretic mobility, it may be helpful to switch the two plots and show the extended plot in a larger format, leaving more space for the reduced plot at the bottom right.

Figure 6: The collective data from all previous experiment is in my opinion better presented in a summarizing heatmap with hierarchical clustering (as in b) and characteristic explanations for the individual groups. The PCA can serve as additional material in the supplement.

Table 2: Instead of 'variation' the word 'effect' is more fitting. Table could be simpler and smaller by dividing boxes diagonally for 'growth' and 'adhesion'/'biofilm' phase.

Discussion

Are there correlations/observations between the measured characteristics and in vivo behavior/physiological capabilities between the different strains?

Other

'homo-aggregation coefficient' are used interchangeably, but rarely include the time variable

The methods state that the homo-aggregation assays were performed in a time-course but the results only report data after 5 h. Are there differences in ACT for the different time points?

line 46: replace '...' with 'etc.'

line 47: change to 'has also shown that even moderate stress can lead to...'

line 145: plural 'were' should be used

Author Response

We sincerely thank Reviewer#2 for her/his positive and relevant comments.

Round 2

Reviewer 1 Report

I am grateful to the Authors to their good work. I am completely satisfied with most of responces. However, I have to retain several of my original comments. I provided mu initial comments (bold) and more detailed comments. In addition, I have one very minor coment on names of bacteria (l. 199 of revised version).

-l. 119. Bf should be B. fragilis.

-Plese, check Your amplification profile. Is 40 cycles reffered to “95 °C for 15 seconds, 60 °C for 1 minute” but not 95 °C 5 min? Where is the elongation phase?

According to MESA BLUE qPCR MasterMix guidelines, 95 °C 5 min are also reffered to initial denaturation, wherease "95 °C for 15 seconds, 60 °C for 1 minute" is reffered to 40 cycles. Please, see the guidelines clearly.

-l. 114-115. Please, provide the reference for “Mueller-Hinton (MH) (1/10) and NaNO3 electrolyte”

I meant the reference to the composition of the medium. Please, rewrite (1/10) to "ten times-diluted"

-l. 114. 1/10 - is it dilution - please, specify.

Please, rewrite (1/10) to "ten times-diluted"

-l. 136. Indicate, how did You determine optical density? In a spectrophotometer? In standard quartz cuvettes?

I saw the response, but there are no corresponding changes in the text. Olease, indicate also name of your spectrophotometer and optical pathlength. It is important for scattered samples, because You cannot compare OD of suspensions on different devices and in different measuring systems.

-l. 139. The expression is not numbered. Why is it in bold?

It is still in bold.

-l. 173. Please, indicate Your light source for fluorescence excitation and the spectral channel for fluorescence detection. Did You use light filters? 

I have seen the response, but there are no corresponding changes in the text.-Please, describe in more detail the procedure of adhesion capacity determination. Did You calculate the cells? How? The same is true about cell number in the biofilms.

The procedure description os OK, but 16S should be 16S rRNA. Please, provide a reference to a protocol for "specific region" (V4/V4-V5 region?) and PCR conditions.

Author Response

We sincerely thank Reviewer#1 for her/his relevant following comments which have allowed us to improve the manuscript.
The manuscript was corrected and improved according to the reviewers’ comments.
Thank you very much for considering this revised manuscript for publication.
